# Temperature-induced amorphization in CaCO₃ at high pressure and implications for recycled CaCO₃ in subduction zones

Mingqiang Hou[1,2,7], Qian Zhang[1,7], Renbiao Tao[3], Hong Liu[4], Yoshio Kono [5,6], Ho-kwang Mao[1,3], Wenge Yang[1], Bin Chen[1] & Yingwei Fei[3]

Calcium carbonate (CaCO₃) significantly affects the properties of upper mantle and plays a key role in deep carbon recycling. However, its phase relations above 3 GPa and 1000 K are controversial. Here we report a reversible temperature-induced aragonite-amorphization transition in CaCO₃ at 3.9–7.5 GPa and temperature above 1000 K. Amorphous CaCO₃ shares a similar structure as liquid CaCO₃ but with much larger C-O and Ca-Ca bond lengths, indicating a lower density and a mechanism of lattice collapse for the temperature-induced amorphous phase. The less dense amorphous phase compared with the liquid provides an explanation for the observed CaCO₃ melting curve overturn at about 6 GPa. Amorphous CaCO₃ is stable at subduction zone conditions and could aid the recycling of carbon to the surface.

[1] Center for High Pressure Science and Technology Advanced Research, 201203 Shanghai, China. [2] The Advanced Light Source, Lawrence Berkeley National Laboratory, Berkeley, CA 94720, USA. [3] Geophysical Laboratory, Carnegie Institution of Washington, Washington, DC 20015, USA. [4] CEA Key Laboratory of Earthquake Prediction, Institute of Earthquake Science, China Earthquake Administration, 100036 Beijing, China. [5] HPCAT, Geophysical Laboratory, Carnegie Institution of Washington, Argonne, IL 60439, USA. [6] Geodynamics Research Center, Ehime University, Matsuyama, Ehime 7908577, Japan. [7] These authors contributed equally: Mingqiang Hou, Qian Zhang. Correspondence and requests for materials should be addressed to M.H. (email: mingqiang.hou@hpstar.ac.cn)

Large amounts of carbon could be introduced into the deep mantle in the form of carbonates (e.g. $CaCO_3$, $MgCO_3$, and $CaMg(CO_3)_2$) by subduction of oceanic crust, constituting an important part of the global carbon cycling. The subducted carbonates pose enormous impacts on the Earth's mantle. Carbonates could drastically reduce melting temperature of peridotite and eclogite[1–3] and are important metasomatic agents with a remarkable wetting capability to impregnate silicate minerals[4–6]. Carbonate melt is an ionic liquid with ultra-low viscosity[7] and is considered responsible for the conductivity anomalies in the oceanic mantle[3,8–10]. In particular, $CaCO_3$ acts as a carrier to transport carbon into deep Earth from surface and transfer it back via volcano eruption[11,12]. $CaCO_3$, together with $CaMg(CO_3)_2$, is also a possible calcium source for the formation of perovskite-structured $CaSiO_3$ observed in deep diamond inclusions, providing insights into the recycling of oceanic crust in the deep mantle[13]. Although $CaCO_3$ was reported to react with enstatite at upper mantle conditions to form dolomite[14], diamond inclusions and exhumed ancient subduction-zone rocks evidently show that $CaCO_3$ can survive to depths of at least the topmost lower mantle[12,15,16].

Extensive studies on phase transitions of $CaCO_3$ at high pressure and temperature have been conducted[17–22]. Rhombohedral calcite (calcite-I) is stable at ambient conditions and transforms to a monoclinic ($P2_1/c$) structure at about 1.5 GPa upon room temperature compression. It further transforms to triclinic calcite-III at about 2 GPa[23]. Upon heating, calcite-III transforms to aragonite[20,24]. However, the phase diagram of $CaCO_3$ above 3 GPa and 1000 K is still controversial. A disordered calcite crystal phase (resembling calcite-IV or -V) was reported according to energy-dispersive X-ray diffraction (EDXRD) patterns[20], while Litasov et al.[25] suggested that it might be a new phase. Both studies did not provide any detailed structure information of the high-$PT$ phase. Coincidently, the $P$–$T$ conditions of the unsolved phase are in accord with the dissolution conditions of $CaCO_3$ from subducted slabs[11]. However, the large-scale dissolution of $CaCO_3$ during subduction remains unclear[11,12]. Therefore, exploring properties of the unsolved phase is essential to unveil this mechanism.

This study aims to clarify the nature of the transition from aragonite to the disordered crystal phase or new phase mentioned in Suito et al.[20] and Litasov et al.[25], respectively, and provide its structure information by in situ measurements up to 7.5 GPa and 1723 K using Paris–Edinburgh press techniques coupled with EDXRD. We report a reversible temperature-induced amorphization in aragonite. The structure measurements of amorphous $CaCO_3$ indicate that the amorphous phase is one of the lightest materials in subducting slabs. Correlations between properties of the amorphous phase and the mechanism of large-scale $CaCO_3$ dissolution in subduction zone are subsequently presented.

## Results

**Phase transitions in CaCO₃.** Upon compression of $CaCO_3$ in a standard assembly cell for large-volume Paris–Edinburgh press (Fig. 1) at room temperature, we observed the transition from the rhombohedral calcite (calcite-I) to the monoclinic phase (calcite-II) between 1.4 and 1.9 GPa. The calcite-III phase appeared at 2.5 GPa (Fig. 2). The observed transitions are consistent with prior determinations of the phase boundaries[17,18,20]. Upon heating at 4.8 GPa, we confirmed the transition to aragonite at 473 K, whose XRD pattern can be indexed with an orthorhombic structure (Fig. 3). The transition temperature is in a good agreement with previous observations[20,24].

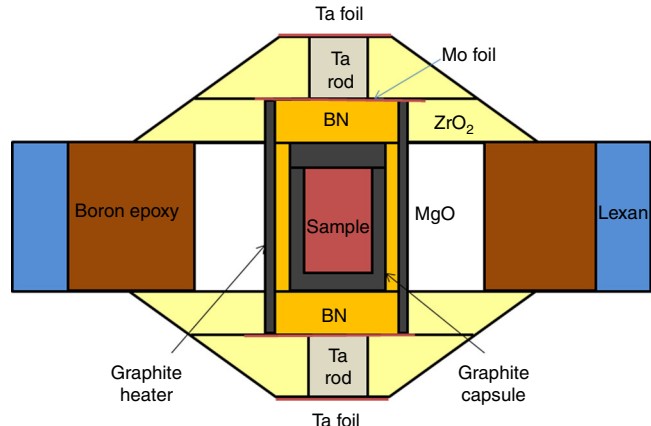

**Fig. 1** The cell assembly for Paris–Edinburgh press used in phase and structure measurements of $CaCO_3$. The $CaCO_3$ sample is loaded in a graphite capsule which is isolated from a graphite heater by a BN pressure medium. The graphite heater is connected by Mo foils and Ta rods which serve as electrodes. MgO serves as both pressure calibrate and pressure medium

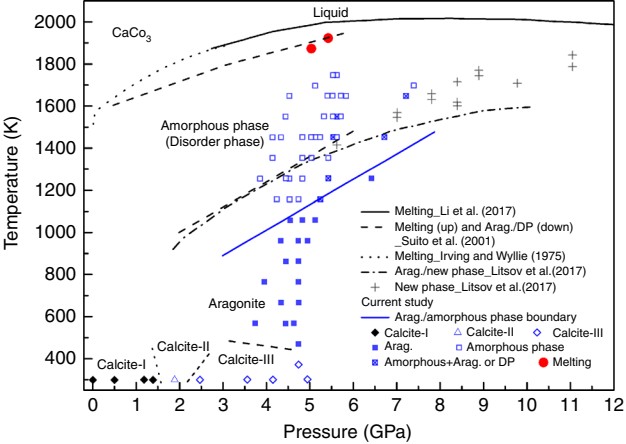

**Fig. 2** Phase diagram of $CaCO_3$ at high pressure and high temperature. Arag. and DP represent aragonite and disorder phase, respectively. The black diamond and colored symbols show current data. The phase boundaries of calcite-I, calcite-III, calcite-II, and aragonite are drawn according to Suito et al.[20]. Other symbols and curves are from Irving and Wyllie[57], Suito et al.[20], Litasov et al.[25], and Li et al.[37]

Upon further heating at 4.8 GPa, we observed the aragonite phase stable up to 1073 K (Fig. 3). During the heating cycle, the recorded diffraction peaks showed some variation in peak intensity due to aragonite grain growth. At 1273 K, the majority of sharp peaks disappeared, and two broad bands emerged at $d$-spacing values of ~1.0 and ~1.3 Å. At higher temperature, only three broad peaks were observed up to 1673 K (Fig. 3). The broad peaks over this temperature range cannot be caused by melting because the temperatures are too low to initiate melting. We also confirmed melting that occurred at temperatures >1873 K by monitoring the movement of Pt spheres in the molten sample. Below the melting temperature, no movement of Pt spheres was observed (Fig. 4). It had been reported that the incorporation of water could significantly reduce the melting temperature of $CaCO_3$ to less than 1127 K[26,27]. However, water (or water source, e.g., $Ca(OH)_2$) should be sealed in Pt (or Au) capsule in large-volume press experiments in order to maintain the equilibrium

reaction. We ran the experiments with pre-dried sample in open capsules and the melting of $CaCO_3$ at relatively low temperature caused by hydration can be ruled out. Furthermore, our experiments with Pt spheres would not support $CaCO_3$ melting at relatively low temperature because the viscosity of the melted $CaCO_3$ is too small to hold the Pt sphere[28] and Pt sphere is expected to drop. On the contrary, we observed the movement of the Pt sphere only when temperature reaches melting >1873 K. The likely explanation for the observed diffraction features is due to a transition to an amorphous phase. Amorphous $CaCO_3$ could be further confirmed by multiangle EDXRD at $2\theta = 3°$, $4°$, $5°$, $7.4°$, $9°$, $12°$, $16°$, and $22°$ (Fig. 5a). No crystal peak of the sample

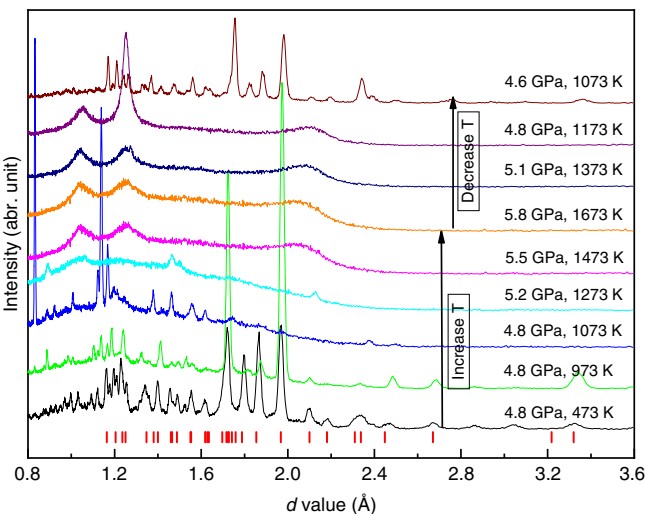

**Fig. 3** Energy-dispersive X-ray diffraction patterns of $CaCO_3$ at high pressure and high temperature. The data were collected at a fixed $2\theta$ of 15° and the energy ($E$) in the $x$ axis was transferred to $d$-value by the equation of $Ed\sin(\theta) = 6.199$ keV·Å. The short red bars denote the peaks of aragonite

was observed at 5.2 GPa and 1773 K in all the EDXRDs. On the other hand, the intensity of the broad peaks is large enough to exclude the possibility of background. More importantly, the broad peaks shift to lower energy when the diffraction angle increases. All the features conclusively verify that $CaCO_3$ becomes an amorphous phase, ruling out the causes by melting, significant grain growth, or single crystal. We subsequently obtained EDXRD patterns during the cooling cycle. The three broad peaks persist to temperatures as low as 1173 K. When the temperature decreased to 1073 K, the sample immediately transformed back to aragonite characterized by its sharp diffraction peaks (Fig. 3). We also observed similar change of the patterns during the heating and cooling cycles in different experiments. These results indicate that the transition between aragonite and the amorphous phase is reversible.

A disordered phase of $CaCO_3$ at 6.16 GPa and 1473 K was reported by Suito et al.[20] The disordered phase resembles calcite-IV or calcite-V with space groups of $R\bar{3}c$ and $R\bar{3}m$, respectively[29], based on 3 diffraction peaks (102), (104), and (110). The diffraction patterns we collected at temperatures >1473 K and 5 GPa are consistent with features of an amorphous phase, and they are distinctively different from those in Suito et al.[20] We also observed some sharp peaks along with the emerging of broad peaks in the temperature range of 1073–1273 K. During ascending temperature, some sharp peaks occasionally persist to temperatures as high as 1673 K and randomly disappeared at a temperature between 1273 and 1673 K in different heating cycles (Fig. 2). Therefore, we cannot rule out the existence of a disordered phase before final amorphization. We note that a weak broad peak at ~56 keV ($d = $~2.1 Å) could be observed at 1473 K in Fig. 4 of Suito et al.[20] The weak feature could be due to relatively short collection time. The $d$-value of the weak peak is consistent with the third broad peak in Fig. 3, indicating that amorphous $CaCO_3$ was also presented in the study of Suito et al.[20] The other two broad peaks shown in Fig. 3 could not be identified in Suito et al.[20], which might be covered by strong graphite peaks at 38 keV.

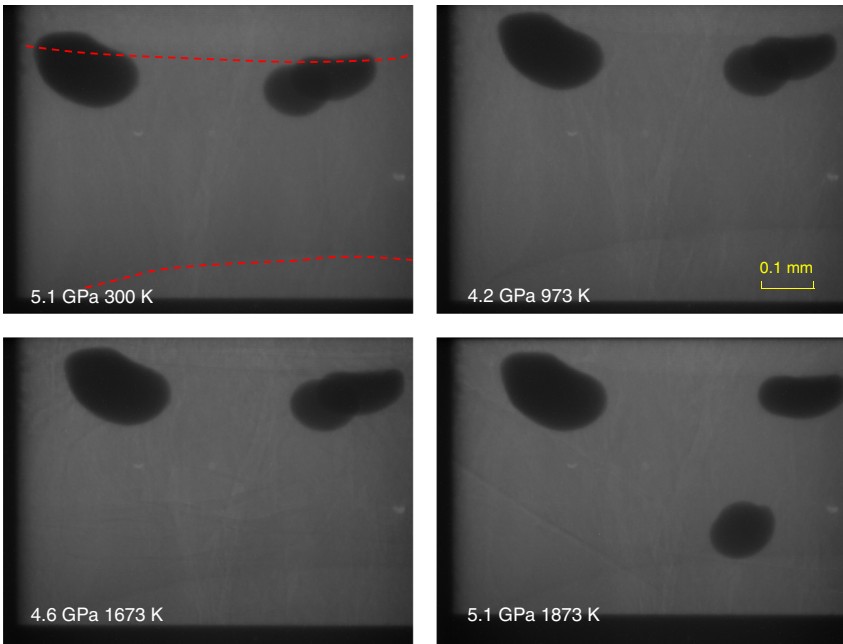

**Fig. 4** Monitoring melting of $CaCO_3$ by movement of Pt spheres in a heating cycle. The two red dash lines denote the interface between sample and graphite capsule. Three Pt balls were loaded in the sample. Two balls were packed near the graphite capsule edge and were stuck. The third Pt ball fell at 1873 K, which meant the melting temperature of $CaCO_3$ was 1873 K at 5.1 GPa

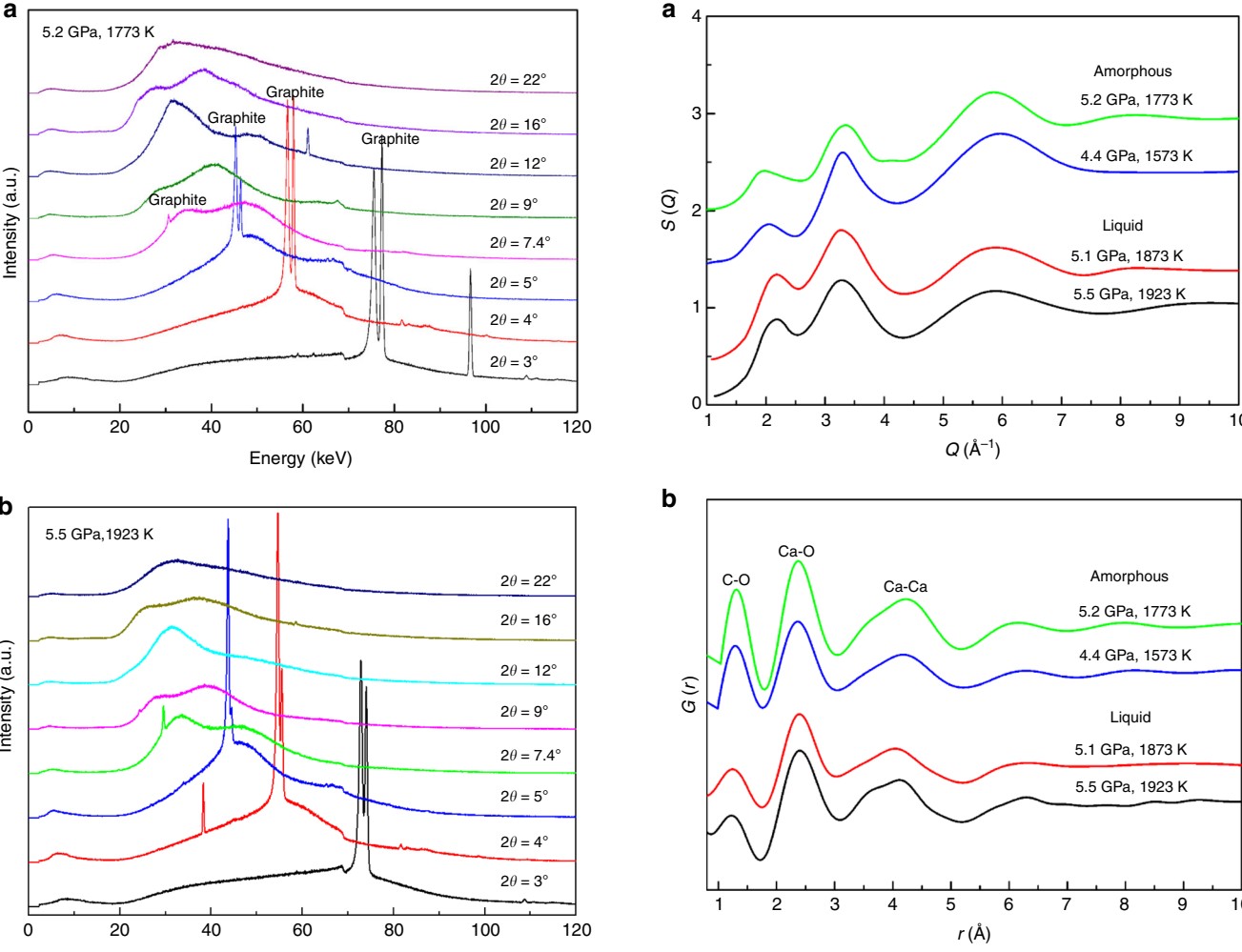

**Fig. 5** Multi-angle energy-dispersive X-ray diffractions of $CaCO_3$. **a** Amorphous $CaCO_3$ at 5.2 GPa and 1773 K; **b** Liquid $CaCO_3$ at 5.5 GPa and 1923 K. The liquid state was determined by the falling of Pt sphere. Two sharp peaks at 61.114 keV ($2\theta = 12°$) and 96.640 keV ($2\theta = 3°$) with $d = 0.970(1)$ Å and $d = 2.450(1)$ Å in **a** are from MgO. The sharp peak at 38.279 keV ($2\theta = 4°$) with $d = 4.640$ (1) Å in **b** may come from other assembly materials

**Fig. 6** Structure information obtained from multi-angle energy-dispersive X-ray diffraction measurements on amorphous $CaCO_3$ at 1573 and 1773 K and liquid $CaCO_3$ at 1873 and 1923 K. **a** Derived structure factor $S(Q)$ of amorphous $CaCO_3$ is compared with that of liquid $CaCO_3$. **b** The corresponding pair distribution function $G(r)$ of amorphous and liquid $CaCO_3$ shows bond length variations for C-O, C-Ca, and Ca-Ca bonds

We conducted a total of seven independent experiments and repeated the experiments to cross the crystal–amorphous phase boundary and characterized this boundary by the diffraction data of the high-*PT* phase. As discussed above, there is some uncertainty regarding when the diffraction peaks of the crystalline phase completely disappear during the heating cycle. However, during descending of the temperature, we consistently observed amorphous $CaCO_3$ without crystalline diffraction peaks until it transformed to aragonite at a definitive temperature. We determine the phase boundary in Fig. 2 via EDXRD data at descending temperature. The boundary determined in this study is about $200 \pm 100$ K lower than that reported by Suito et al.[20] and Litasov et al.[25]. Our boundary may be considered as a low bound for the transition.

**Structure information of the amorphous phase and liquid $CaCO_3$.** Broad diffraction peaks, observed at temperatures between 1473 and 1673 K (Fig. 3), are consistent with diffraction signal from an amorphous phase. In order to confirm the amorphization of $CaCO_3$ at high temperature and to understand the differences between amorphous and liquid $CaCO_3$, we conducted multiangle EDXRD measurements at diffraction angles of 3–22° up to 1923 K at 3.9–7.5 GPa. Figure 5 presents the typical multiangle EDXRDs of amorphous and liquid $CaCO_3$. One or two broad peaks were observed in the multiangle EDXRD at each $2\theta$ (Fig. 5). As the $2\theta$ increases, the broad peaks shifted to lower energy and new broad peak may appear at higher energy. From the multiangle EDXRD, we can get the structure factor [$S(Q)$] and real-space pair distribution function [$G(r)$][28].

Figure 6a shows the representative structure factor [$S(Q)$] of amorphous and liquid $CaCO_3$ up to a $Q$ value of 10 Å$^{-1}$. The $S(Q)$ patterns for the amorphous solid phase (1573 and 1773 K) and liquid (1873 and 1923 K) share similar features. The first sharp diffraction peak (FSDP) in $S(Q)$ reflects the intermediate range ordering in network forming liquid or glass[30]. The intensity of FSDP for amorphous $CaCO_3$ is slightly smaller than that of liquid $CaCO_3$. However, the amorphous phase has a broader FSDP, indicating different intermediate range orderings in amorphous and liquid $CaCO_3$.

**Table 1 The bond lengths of C-O, Ca-O, and Ca-Ca in aragonite, amorphous, and liquid CaCO₃**

|  | P (GPa) | T (K) | C-O (Å) | Ca-O (Å) | Ca-Ca(Å)[b] |
|---|---|---|---|---|---|
| Liquid | 5.5 | 1923 | 1.193(3) | 2.444(3) | 4.114(2) |
|  | 5.1 | 1873 | 1.210(2) | 2.439(2) | 4.040(1) |
| Amorphous | 5.2 | 1773 | 1.319(1) | 2.406(2) | 4.232(2) |
|  | 4.4 | 1573 | 1.304(1) | 2.381(1) | 4.177(1) |
| Aragonite[a] | 5.2 | 1773 | 1.288(1) | 2.437(1) | 3.922(1) |
|  | 4.4 | 1573 | 1.287(1) | 2.435(1) | 3.976(1) |

[a]The bond lengths of aragonite-CaCO₃ were calculated according to Litasov et al.[25] and Negro and Ungaretti[33]
[b]The Ca-Ca bond lengths for amorphous phase and liquid phase are difficult to be calculated because of the asymmetry of the peak. The values are the positions of the topmost of the Ca-Ca peaks

Fourier transformation of $S(Q)$ yielded the real-space pair distribution function, $G(r)$ (Fig. 6b)[31]. The $G(r)$ patterns of liquid CaCO₃ are similar to those reported in previous studies[7,32]. The amorphous CaCO₃ has a similar $G(r)$ pattern to that of the liquid CaCO₃. However, differences can be observed in the bond lengths of C-O, Ca-O, and Ca-Ca, represented by the first, second, and third peaks, respectively (Table 1). The bond lengths of amorphous and liquid CaCO₃ were calculated from peaks in $G(r)$ patterns using Gaussian fitting and those of aragonite were estimated according to its thermal equation of state and crystal structure[25,33]. The C-O bond length of the amorphous phase is almost same as aragonite, while it is much larger than that of the liquid phase. The Ca-O bond length of the amorphous phase is identical to those of liquid CaCO₃ and aragonite. It is difficult to get the Ca-Ca bond length because of the asymmetry of the Ca-Ca peak. However, we can qualitatively compare the differences by the topmost position (TP) of asymmetric Ca-Ca peaks. The Ca-Ca TPs of liquid and aragonite are ~6.3% and ~7.9% smaller than that of the amorphous phase, respectively. These data indicate that the amorphous phase at 1573 and 1773 K has basically the same structure as liquid but with different C-O and Ca-Ca bond lengths.

Although it is difficult to obtain an accurate density from the diffraction data of liquid or amorphous phase alone, the measurements provide qualitative comparison of densities for different phases because the density is proportional to the FSDP position[34]. The center of the FSDP of amorphous CaCO₃ is ~2.00 Å⁻¹, while that of the liquid phase is ~2.15 Å⁻¹. The former is about 7% smaller, which is significantly larger than the uncertainty (<1%)[35] in determining the center of the FSDP. Our diffraction data indicate that the density of the amorphous CaCO₃ is smaller than that of the liquid phase based on the extracted structure factor of the amorphous and liquid phases. This can be further confirmed by the comparisons of the bond lengths in Table 1. Zhang and Liu[36] theoretically calculated the densities of aragonite and liquid CaCO₃ and found that the latter is ~13% smaller. In Table 1, the Ca-Ca bond length of liquid CaCO₃ is much larger than that of aragonite, while their Ca-O bond lengths are comparable. From the density comparison of liquid CaCO₃ and aragonite, we can conclude that the Ca-Ca matrix determines the density, while Ca-O and C-O are the second factor to affect density. Therefore, amorphous CaCO₃ is less dense than its liquid counterpart, because the Ca-Ca bond length of amorphous CaCO₃ is larger. The Clapeyron relation $\left(\frac{dT}{dP} = \frac{\Delta V}{\Delta S}\right)$ describes the relation between phase boundary slope $\left(\frac{dT}{dP}\right)$ and volume change ($\Delta V$), where $\Delta S$ is entropy change. Since amorphous CaCO₃ is less dense than liquid CaCO₃, $\Delta V$ would be negative, leading $\frac{dT}{dP}$ to a negative value. This inference

is consistent with the observed melting curve overturn at about 6 GPa[20,37].

**Mechanism**. Pressure-induced amorphization is fundamentally interesting in physics, chemistry, material, geoscience, and industrial applications[38,39]. On the other hand, temperature is mainly treated as a crystallization stabilizer. Only hexagonal ice, SiO₂-stishovite, and zeolites were reported to transform to an amorphous phase upon heating[40–42]. The temperature-induced amorphization (TIA) in CaCO₃ is more intriguing because (1) amorphous CaCO₃ is a stable phase and does not recrystallize before melting; and (2) the aragonite–amorphous phase transition is reversible. We observed direct melting from the amorphous phase without recrystallization in CaCO₃, which behaves totally different from the crystal–amorphous phase–recrystallization in the meta-stable stishovite at ambient pressure. We also noticed that the amorphous CaCO₃ with a relatively larger volume (the sample size is) could be stabilized at high pressure and high temperature compared with the metastable small amorphous CaCO₃ particle (~1 μm) intergrown with aragonite at ambient conditions[43,44], indicating that amorphous CaCO₃ would exist in volumetrically large size in subduction zone.

The reported mechanisms for TIA include chemical disorder, thermodynamic melting followed by immediate vitrification, and mechanical collapse[42,45]. Chemical disorder is considered in zeolite where Al substitutes Si in TO₄ framework[45], whereas it can be excluded in the case of amorphous CaCO₃ because there is no another element to substitute carbon in the framework. Thermodynamic melting followed by immediate vitrification was favored in amorphization of hexagonal ice[42,45]. Taking into account the melting temperature determined by Suito et al.[20] and Li et al.[37], the observed amorphization of CaCO₃ occurs at much too low temperature (~800 K lower than the melting temperature) to be explained by vitrification. Consequently, mechanical collapse is the most plausible mechanism for the TIA in CaCO₃. The crystal structure might be destroyed by the high mobility of the Ca²⁺ cation and the CO₃²⁻ anion in the amorphous phase. This is consistent with the moderate decrease of the measured resistance across the aragonite to amorphous phase transition below the melting temperature[37]. The increase of the Ca-Ca bond length of the amorphous phase compared to that of aragonite also supports the increased mobility of Ca²⁺ cation that could eventually lead to the destruction of the Ca²⁺ lattices to form the amorphous phase.

**Discussion**

The observed amorphization in CaCO₃ occurs at a relatively low temperature, comparable to the conditions relevant to subduction zone environments. At depth of 30–50 km, calcite will transform to aragonite and further to amorphous phase around 70–120 km (Fig. 7). The bond length comparison and the negative slope of the melting curve support that amorphous CaCO₃ is less dense than the liquid counterpart. Most recently, Hudspeth et al.[32] obtained the densities of liquid CaCO₃ based on structure measurements, which range from 2.4 to 2.6 g cm⁻³ in the pressure range of 4.7–8.7 GPa at ~2000 K. In contrast, the densities of aragonite are 2.9–3.1 g cm⁻³ at these $P$–$T$ conditions according to Litasov et al.[25]. Liquid CaCO₃ is ~16% lighter than aragonite. The density reduction of liquid CaCO₃ compared with aragonite is also supported by first principle calculations. Zhang and Liu[36] calculated the densities of aragonite and liquid CaCO₃ and found the latter is ~13% smaller. Since the amorphous CaCO₃ is less dense than the liquid CaCO₃, it would be at least 16% smaller than that of aragonite. The density profile of aragonite[25] is comparable to MgCO₃[46], (Fe₀.₆₅Mg₀.₃₃Mn₀.₀₂)CO₃[21], basalt

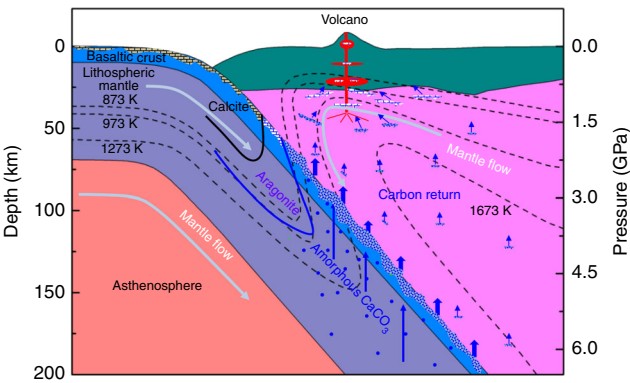

**Fig. 7** Phases of CaCO$_3$ at upper mantle conditions and the schematic for deep carbon recycle. The geotherms follow Gerya and Yuen[58] and Penniston-Dorland et al.[59]. The blue dot and blue shaded area in subducting slabs represent the amorphous CaCO$_3$ and concentration area of amorphous CaCO$_3$, respectively. The blue shaded particles in mantle wedge are escaped amorphous CaCO$_3$ through percolation. At the top of the asthenosphere in mantle wedge, the amorphous CaCO$_3$ would transform back to aragonite

melt[47], and PREM[48]. Our results imply that the amorphous CaCO$_3$ is one of the lightest materials in subduction zone assemblages.

The phase boundary of aragonite and amorphous CaCO$_3$ intersects the melting curve at ~13 GPa, which corresponds to a depth of 400 km. In the depth between 70 and 400 km of subducted slabs, the most plausible form of CaCO$_3$ is amorphous phase according to the phase diagram in Fig. 2. The density difference between the amorphous CaCO$_3$ and other subduction zone assemblages indicates that the amorphous CaCO$_3$ would possess an ultra-high buoyancy. On the other hand, amorphous CaCO$_3$ is a weak solid with shear modulus of 14 GPa at ambient conditions[49], which is only ~17% of olivine[50] and ~41% of aragonite[51]. The ultra-high buoyant and soft amorphous CaCO$_3$ has important implications for carbon recycling in subduction zones. Indeed, the majority of CaCO$_3$ would likely escape the downgoing slabs and ascend to magmas in the overlying mantle through dissolution[11], rather than subducting to the deep mantle (Fig. 7). Decarbonization in fluids from the shallow subducting slabs only accounts for a portion of CO$_2$ released in volcanic gas emissions[52–54]. CaCO$_3$ has been considered to fill the gap between the CO$_2$ outputs via volcanism and inputs in subducting slabs[12]. However, the mechanism for a large-scale CaCO$_3$ dissolution from the slab is unknown[11]. We propose a novel regime of carbon input where CaCO$_3$ readily concentrates through percolation of the amorphous phase and migrates from the subducting slabs to the mantle wedge. The migration of CaCO$_3$ in the mantle wedge could be complex depending on the thermal structure of the slab and mantle wedge. Because the mantle wedge is hotter than the subducted slabs, the percolated amorphous CaCO$_3$ from the subducted slabs would be further stabilized and migrate faster when CaCO$_3$ enters the mantle wedge (Fig. 7). On the other hand, the higher temperature in the mantle wedge would dilate the amorphous CaCO$_3$, so as to increase its buoyancy. In the shallow part of the mantle wedge, the temperature decreases as depth becomes smaller. The buoyancy of the amorphous CaCO$_3$ would be reduced and the migration of the amorphous CaCO$_3$ would be slower. As the amorphous CaCO$_3$ migrates to a depth where the temperature is lower than the phase boundary between aragonite and the amorphous phase, it would transform back to aragonite. Assimilation of the amorphous CaCO$_3$ into arc magmas could serve as an important CO$_2$ source

for volcanic eruptions. Additionally, this mechanism may lead to the formation of calcite-rich carbonatites[12,55].

## Methods
**Experimental set-up.** The EDXRD measurements on CaCO$_3$ at high pressure and temperature were conducted using a large-volume Paris–Edinburgh press at 16-BMB synchrotron beamline at the Advanced Photon Source, Argonne National Laboratory. Two cup-shaped tungsten carbide anvils were used to generate high pressures. The starting material, CaCO$_3$ (purity 99.99%, Alfa Aesar company) dried for 24 h in an oven at 110 °C, was loaded into a graphite capsule (Fig. 1). A boron nitride sleeve was used to separate the graphite heater from the capsule. An MgO sleeve outside the graphite heater served as both pressure medium and pressure marker. Pressure was determined from the XRD data of MgO[56] at a fixed angle (2$\theta$) of 15° using a Ge solid-state detector. Temperature was determined according to the calibrated temperature–power relationship[28]. The pressure and temperature uncertainties are 0.3 GPa and 100 K, respectively. Pt spheres of 80–100 μm in diameter were loaded in the sample chamber to monitor the melting of sample.

**Structure measurements.** The structure measurements of the solid and liquid CaCO$_3$ were carried out via multiangle EDXRD fixed at 2$\theta$s of 3°, 4°, 5°, 7.4°, 9°, 12°, 16°, and 22°. The collection time of EDXRD at each angle is 30 min. Structure factor [$S(Q)$] and real-space pair distribution function [$G(r)$] are obtained as follows,

$$S(Q) = \frac{I_m(2\theta, E) - S'(2\theta)I_{p,eff}(E)[f^2(2\theta, E) + I_{inc}(2\theta, E)]}{S'(2\theta)I_{p,eff}(E)f^2(2\theta, E)^2} \quad (1)$$

$$I_{p,eff}(E) = A(E)C(E)I_p(E) \quad (2)$$

$$G(r) = \frac{2}{\pi} \int_{Q_{min}}^{Q_{max}} Q[S(Q) - 1]\sin(Qr)dQ \quad (3)$$

where $Q = \frac{4\pi E}{12.3984}\sin(\theta)$, $I_m(2\theta, E)$ is the observed energy EDXRD spectrum at a given 2$\theta$, $S'(2\theta)$ is proportional to $\cos^2(2\theta)$, $I_p(E)$ is the primary white beam profile, $A(E)$ is the X-ray attenuation, $C(E)$ is any other energy-dependent term, $I_{inc}(2\theta, E)$ is incoherent scattering from the sample, $f^2(2\theta, E)$ can be calculated according to parameters in International Tables for X-ray Crystallography (Ed. Ibers and Hamilton). The detailed data analysis method can be found in Kono et al.[28].

## Data availability
The data that support the findings of this study are available from the corresponding author upon request.

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

## Acknowledgements

The authors would like to thank Dr. Jung-Fu Lin, Dr. Qishi Zeng, and Dr. Howard Sheng for their suggestions in understanding the TIA in CaCO₃. We also thank Dr. Martin Kunz and Dr. Amanda Lindoo for reviewing the manuscript. Special thanks to Dr. Huiyang Gou and Dr. Zhisheng Zhao for their contributions in analyzing the mechanism of TIA in CaCO₃. This work was supported by NSAF (Grant No. U1530402) and NSFC (Grant No. 41174071). The experiments were performed at HPCAT (Sector 16), Advanced Photon Source (APS), Argonne National Laboratory. HPCAT operation is supported by DOE-NNSA under Award No. DE-NA0001974, with partial instrumentation funding from NSF. The Advanced Photon Source is a U.S. Department of Energy (DOE) Office of Science User Facility operated for the DOE Office of Science by Argonne National Laboratory under Contract No. DE-AC02–06CH11357. Y.K. acknowledges the support of DOE-BES/DMSE under Award DE-FG02–99ER45775. Y.F acknowledges the support of Carnegie Institution of Washington and NSF grant (EAR-1619868).

## Author contributions

M.H. and Y.F. proposed the project. M.H., Y.K. and Q.Z. conducted the experiments. Q.Z. analyzed the data and Y.K. checked the data analysis. M.H., R.T. and H.L. analyzed the mechanism of the temperature-induced amorphization. W.Y., B.C. and H.-k.M. proposed the structure measurements and discussed the mechanism. M.H., Y.F. and Y.K. wrote the manuscript. All the authors commented on the manuscript.

## Additional information

**Competing interests:** The authors declare no competing interests.

