## [Peer Review File · Nature Communications]

Reviewers' comments:

Reviewer #1 (Remarks to the Author):

I think that the work opens new scenarios about the meaning of carbonate in the mantle and the evidence of an amorphous CaCO₃ phase in my opinion is convincing. So I consider that the work should be published only under minor revisions that I propose here below.

Line 42-45. Just a line about the possibility that CaCO₃ also serves as possible source of calcium found in CaSiO₃ compounds as inclusions in diamonds would just strength the importance of the work (see Nestola et al. 2018, Nature, v. 555, p. 237.; Anzolini et al. 2018, American Mineralogist, 103, 69-74). Indeed, this work could really give an important help to understand the calcium presence at very great depths considering that the CaSiO₃ perovskite in Nestola et al. (2018) was placed at about 780 km depth. I must remark that further two CaSiO₃ real perovskites are now found in other diamonds but these data are not yet published. All these data indicate lower mantle conditions. Of course, we must remember that at pressure of about 8-9 GPa the CaSiO₃ walstromite phase (now called breyite) found in natural diamonds is what we think could be transformed to perovskite. So CaCO₃ could provide calcium to this walstromite phase and then who knows....

Some works would indicate that the main Ca-transport to great depths could be due to garnet (majoritic garnet) which destabilizes in the lower mantle. However, phases like jeffbenite (inclusions in diamonds with the same stoichiometry of garnet, Nestola et al. 2016, Mineralogical Magazine) definitively show that garnet compositions at great depths is practically without calcium....

Line 83-92. Although I don't have doubts that this information is very important, I wonder if it's the case to leave them here in this part of the text or to shift it somewhere else. I leave the decision to the Editor.

Line 108. A minor correction, I would not use "cann't" but simply "can not" or just "cannot". Same applies at line 114 and 142 with "couldn't" and 210 with "doesn't".

Reference list. The reference number 33 has a mistake. The authors are Dal Negro A. (without T) and Ungaretti (adding the final "i"). Same in the note of Table 1.

Reviewer #2 (Remarks to the Author):

The paper reports new investigations of the phase diagram of CaCO₃ of 1900 K and 6 GPa and discusses implications for the recycling of carbon in the Earth's interior. The most notable observation is the reversible temperature-induced amorphization of aragonite before melting takes place. Based on the pair-distribution functions of the liquids and the new amorphous phase, the authors suggest that the amorphous phase is less dense than the liquid, even though numerical constraints for the density of none of the phases are provided. The authors conclude that this new amorphous phase may play a key role in controlling the input of carbon into the mantle wedge. While I think that the experimental results are quite sound, the implications for the recycling of carbon remains poorly supported by arguments.

Below there is a list of specific comments:

1) It is not clear from the experimental details if the amorphous phase was confirmed in repeated experiments or if the reported data is the result of only one experimental run. This should be clarified.

2) In subducted lithologies, calcite/aragonite may not be stable as an individual phase and carbon will be rather stored in dolomite or dolomite-magnesite solid solutions (see Dasgupta 2013 RIMG volume). Therefore, the CaCO₃ amorphous phase may not be a relevant phase in complex

subducted lithologies and its role in the recycling of carbon rather limited. Is there evidence (experimental or theoretical) for temperature-induced amorphization in dolomite?

3) Lines 265-267: I think this is a rather simplistic view of the recycling process. Even if the amorphous phase would percolate from the slab to the mantle wedge, this does not imply that it would be stable at the higher temperatures of the mantle. The survival of aragonite as an amorphous phase to the depths and conditions of the arc magma source is very unlikely. This paragraph should be modified.

4) Finally, I find the paper poorly written, some sentences seem more appropriate for lab notes than for a manuscript (e.g. lines 148-149 among many other). The writing style should be significantly improved before the paper can be considered for publication.

There are in addition a number of typos, omissions and grammatical mistakes that should be corrected in a revised version:

Minor comments:

Line 65-66: Verb is missing in this sentence

Line 134: "distinctly different" is redundant and should be corrected by "significantly different" or similar

Line 195: The should be the Lines 221, 224, 227: vitrification instead of vetrification

Line 225: Taking into account.....

Line 512: geotherms instead of Geotherms

Figures and tables:

Figure 2, legend: the dashed and dot-dashed lines used to represent literature data are difficult to distinguish. I would suggest using different colors or other symbols.

Reviewer #3 (Remarks to the Author):

The authors present a re-investigation of the CaCO_3 polymorphs at simultaneous high P and T to unravel controversies still existing on the stability of the highP-T polymorphs above 3GPa and 1000K.

The study is performed using state of the art equipment and the interpretation is carried out carefully. However, large part of the implications is hang on to the contrast in densities that is about 7% (e.g. difference in the center of the FSDP of about 0.15angstrom out of 2 angstrom). I wonder what would it be the uncertainty on such determination. Is it the difference higher than 1 or 2 estimated uncertainties? One more question is that relative to the bond lengths reported in table 1 is this is another strong constraint. As stated by the authors Ca-Ca bonds in amorphous and liquid are difficult to determine and yet the estimated uncertainties are basically the same as those for aragonite. Is the fit goodness accounted in the uncertainties reported?

If these issues can be cleared out then the manuscript and its broad conclusions holds and I therefore recommend publishing as is in its current state. On the contrary I would at the very least recommend the authors to emphasize more the limitations of such method.

Point-to-point respond

Reviewer #1 (Remarks to the Author):

I think that the work opens new scenarios about the meaning of carbonate in the mantle and the evidence of an amorphous CaCO₃ phase in my opinion is convincing. So I consider that the work should be published only under minor revisions that I propose here below.

Line 42-45. Just a line about the possibility that CaCO₃ also serves as possible source of calcium found in CaSiO₃ compounds as inclusions in diamonds would just strength the importance of the work (see Nestola et al. 2018, Nature, v. 555, p. 237.; Anzolini et al. 2018, American Mineralogist, 103, 69-74). Indeed, this work could really give an important help to understand the calcium presence at very great depths considering that the CaSiO₃ perovskite in Nestola et al. (2018) was placed at about 780 km depth. I must remark that further two CaSiO₃ real perovskites are now found in other diamonds but these data are not yet published. All these data indicate lower mantle conditions. Of course, we must remember that at pressure of about 8-9 GPa the CaSiO₃ walstromite phase (now called breyite) found in natural diamonds is what we think could be transformed to perovskite. So CaCO₃ could provide calcium to this walstromite phase and then who knows....

Some works would indicate that the main Ca-transport to great depths could be due to garnet (majoritic garnet) which destabilizes in the lower mantle. However, phases like jeffbenite (inclusions in diamonds with the same stoichiometry of garnet, Nestola et al. 2016, Mineralogical Magazine) definitively show that garnet compositions at great depths is practically without calcium.....

Line 83-92. Although I don't have doubts that this information is very important, I wonder if it's the case to leave them here in this part of the text or to shift it somewhere else. I leave the decision to the Editor.

Line 108. A minor correction, I would not use "cann't" but simply "can not" or just "cannot". Same applies at line 114 and 142 with "couldn't" and 210 with "doesn't".

Reference list. The reference number 33 has a mistake. The authors are Dal Negro A.

(without T) and Ungaretti (adding the final “i”). Same in the note of Table 1.

Responds to Review #1

Question 1# Just a line about the possibility that CaCO₃ also serves as possible source of calcium found in CaSiO₃ compounds as inclusions in diamonds would just strength the importance of the work (see Nestola et al. 2018, Nature, v. 555, p. 237.; Anzolini et al. 2018, American Mineralogist, 103, 69-74).

Respond: As suggested, we added the reference and a sentence in the main text that CaCO₃ can act as a calcium source for CaSiO₃ (see lines 42-45 in the revised main text).

Question 2# A minor correction, I would not use “cann’t” but simply “can not” or just “cannot”. Same applies at line 114 and 142 with “couldn’t” and 210 with “doesn’t”.

Respond: We edited the text accordingly.

Question 3#

Reference list. The reference number 33 has a mistake. The authors are Dal Negro A. (without T) and Ungaretti (adding the final “i”). Same in the note of Table 1.

Respond: We corrected the mistake.

Reviewer #2:

The paper reports new investigations of the phase diagram of CaCO₃ of 1900 K and 6 GPa and discusses implications for the recycling of carbon in the Earth’s interior. The most notable observation is the reversible temperature-induced amorphization of aragonite before melting takes place. Based on the pair-distribution functions of the liquids and the new amorphous phase, the authors suggest that the amorphous phase is less dense than the liquid, even though numerical constraints for the density of none of the phases are provided. The authors conclude that this new amorphous phase may play a key role in controlling the input of carbon into the mantle wedge. While I think that the experimental results are quite sound, the implications for the recycling of carbon remains poorly supported by arguments.

Below there is a list of specific comments:

1) It is not clear from the experimental details if the amorphous phase was confirmed in repeated experiments or if the reported data is the result of only one experimental run. This should be clarified.

2) In subducted lithologies, calcite/aragonite may not be stable as an individual phase and carbon will be rather stored in dolomite or dolomite-magnesite solid solutions (see Dasgupta 2013 RiMG volume). Therefore, the CaCO₃ amorphous phase may not be a relevant phase in complex subducted lithologies and its role in the recycling of carbon rather limited. Is there evidence (experimental or theoretical) for temperature-induced amorphization in dolomite?

3) Lines 265-267: I think this is a rather simplistic view of the recycling process. Even if the amorphous phase would percolate from the slab to the mantle wedge, this does not imply that it would be stable at the higher temperatures of the mantle. The survival of aragonite as an amorphous phase to the depths and conditions of the arc magma source is very unlikely. This paragraph should be modified.

4) Finally, I find the paper poorly written, some sentences seem more appropriate for lab notes than for a manuscript (e.g. lines 148-149 among many other). The writing style should be significantly improved before the paper can be considered for publication. There are in addition a number of typos, omissions and grammatical mistakes that should be corrected in a revised version:

Minor comments:

Line 65-66: Verb is missing in this sentence

Line 134: “distinctly different” is redundant and should be corrected by “significantly different” or similar

Line 195: The should be the

Lines 221, 224, 227: vitrification instead of vetrification

Line 225: Taking into account.....

Line 512: geotherms instead of Geotherms

Figures and tables:

Figure 2, legend: the dashed and dot-dashed lines used to represent literature data

are difficult to distinguish. I would suggest using different colors or other symbols.

Question 1# It is not clear from the experimental details if the amorphous phase was confirmed in repeated experiments or if the reported data is the result of only one experimental run. This should be clarified.

Respond: We conducted a total of 7 experiments and observed temperature-induced amorphous CaCO₃ at different temperatures (summarized in Fig. 2). We emphasized this in the revision. Figure 3 shows a typical evolution of the X-ray diffraction patterns in a temperature cycle. We observed similar change of the patterns in different experiments. The structure information obtained from multi-angle EDXRD measurements on amorphous CaCO₃ at 1573 K and 1773 K (Fig. 6) represent the best structure factor data. We have also collected data at P-T conditions in Fig. 2, but the data quality is less ideal for extracting structure information. The following figures are two of them (4.9 GPa-1673 K and 4.2 GPa- 1473 K).

Question 2# In subducted lithologies, calcite/aragonite may not be stable as an individual phase and carbon will be rather stored in dolomite or dolomite-magnesite solid solutions (see Dasgupta 2013 RiMG volume). Therefore, the CaCO₃ amorphous phase may not be a relevant phase in complex subducted lithologies and its role in the recycling of carbon rather limited. Is there evidence (experimental or theoretical) for temperature-induced amorphization in dolomite?

Respond: It is a main concern whether CaCO₃ is stable in subducted lithologies. Indeed there are reports that CaCO₃ would react with enstatite at upper mantle

conditions to form dolomite. But diamond inclusions and exhumed ancient subduction-zone rocks evidently show CaCO_3 can survive to depths of at least the topmost lower mantle. We have emphasized this in line 45-47 of the revised main text.

We have not conducted any experiment on possible temperature-induced amorphization in dolomite yet, but this will be our future research project.

Question 3# Lines 265-267: I think this is a rather simplistic view of the recycling process. Even if the amorphous phase would percolate from the slab to the mantle wedge, this does not imply that it would be stable at the higher temperatures of the mantle. The survival of aragonite as an amorphous phase to the depths and conditions of the arc magma source is very unlikely. This paragraph should be modified.

Respond: We agree that the migration of CaCO_3 in the slab and mantle wedge could be complex. We added more discussion on the recycling processes in the revision (see lines 272-281). We would like to emphasize the unique properties of the amorphous phase such as low-density and stability at higher temperature. Because of the thermal structure of the mantle wedge next to the subducted slabs, the amorphous CaCO_3 would become more stable when CaCO_3 percolates from the subducted slabs to the mantle wedge. The ultra-high buoyant and soft amorphous CaCO_3 could significantly enhance carbon mobility in the slab and mantle wedge.

Question 4# Finally, I find the paper poorly written, some sentences seem more appropriate for lab notes than for a manuscript (e.g. lines 148-149 among many other). The writing style should be significantly improved before the paper can be considered for publication. There are in addition a number of typos, omissions and grammatical mistakes that should be corrected in a revised version:

Minor comments:

Line 65-66: Verb is missing in this sentence

Line 134: “distinctly different” is redundant and should be corrected by “significantly different” or similar

Line 195: The should be the

Lines 221, 224, 227: vitrification instead of vetrification

Line 225: Taking into account.....

Line 512: geotherms instead of Geotherms

Figures and tables:

Figure 2, legend: the dashed and dot-dashed lines used to represent literature data are difficult to distinguish. I would suggest using different colors or other symbols.

Respond: Thank you for the specific comments. Additional effort has been made to improve the presentation of the manuscript (see lines 136-159).

Reviewer #3 (Remarks to the Author):

The authors present a re-inveistigation of the CaCO₃ polymorphs at simultaneous high P and T to unravel controversies still exsiting on the stability of the high P-T polymorphs above 3GPa and 1000K. The study is performed using state of the art equipment and the interpretation is carried out carefully. However, large part of the implications is hang on to the contrast in densities that is about 7% (e.g. difference in the center of the FSDP of about 0.15angstrom out of 2 angstrom). I wonder what would it be the uncertainty on such determination. Is it the difference higher that 1 or 2 estimated uncertainties? One more question is that relative to the bond lengths reported in table 1 s this is another strong constraint. As stated by the authors Ca-Ca bonds in amorphous and liquid are difficult to determine and yet the estimated uncertainties are basically the same as those for aragonite. It the fit goodness accounted in the uncertainties reported?

If these issues can be cleared out then the manuscript and its broad conclusions holds and I therefore recommend publishing as is in its current state. On the contrary I would at the very least recommend the authors to emphasize more the limitations of such method.

Question 1#:

I wonder what would it be the uncertainty on such determination. Is it the difference higher that 1 or 2 estimated uncertainties? One more question is that relative to the bond lengths reported in table 1 s this is another strong constraint. As stated by the

authors Ca-Ca bonds in amorphous and liquid are difficult to determine and yet the estimated uncertainties are basically the same as those for aragonite. It the fit goodness accounted in the uncertainties reported?

Respond: The uncertainty in determine the Q of the first sharp diffraction peak (FSDP) in $S(Q)$ is less than 1%, while difference in Q values between the amorphous phase and liquid is ~7%, sufficiently larger than the uncertainty. We have added a sentence on the uncertainty in the revision.

We only listed the values from the positions of the topmost of the Ca-Ca peaks in Table 1 for a qualitative comparison. The cited uncertainty, derived from the peak position fit only, does not represent the uncertainty in the bond length. Our conclusion that the amorphous CaCO_3 is lighter than its liquid counterpart is supported not only by the qualitative comparison of the Ca-Ca bond length, but also by the comparisons of Q values of FSDP and the Clapeyron relation ($\frac{dT}{dP} = \frac{\Delta V}{\Delta S}$) of the melting curve.

REVIEWERS' COMMENTS:

Reviewer #2 (Remarks to the Author):

The authors have addressed my previous suggestions. However, while I still consider that the implications for the deep carbon cycle are not very clear, the identification of the new amorphous phase in CaCO₃ is of sufficient relevance to justify the publication of the manuscript in Nature Communications.

I have a final suggestion for improving the title that I think it would read better as: "Temperature-induced amorphization in CaCO₃ at high pressure: implication for recycled CaCO₃ in subduction zones"

After that, I will recommend publication of the manuscript in the current form.

Reviewer #3 (Remarks to the Author):

The authors have properly addresses all the comments from the reviewer and the manuscript has been now improved. I therefore recommend for publication in Nature Communication in its present state